# Generative AI extracts ecological meaning from the complex three dimensional shapes of bird bills

**Russell Dinnage** [1,2,3]*, **Marian Kleineberg**[4,5]

**1** Department of Biological Sciences, Institute of Environment, Florida International University, Miami, Florida, United States of America, **2** Institute for Applied Ecology, University of Canberra, Canberra, Australia, **3** Macroecology and Macroevolution Group, Australian National University, Canberra, Australia, **4** Advanced Technology Group, Sony Playstation, London, United Kingdom, **5** Technische Universität Dortmund University, Dortmund, Germany

* rdinnage@fiu.edu

## Abstract

Data on the three dimensional shape of organismal morphology is becoming increasingly available, and forms part of a new revolution in high-throughput phenomics that promises to help understand ecological and evolutionary processes that influence phenotypes at unprecedented scales. However, in order to meet the potential of this revolution we need new data analysis tools to deal with the complexity and heterogeneity of large-scale phenotypic data such as 3D shapes. In this study we explore the potential of generative Artificial Intelligence to help organize and extract meaning from complex 3D data. Specifically, we train a deep representational learning method known as DeepSDF on a dataset of 3D scans of the bills of 2,020 bird species. The model is designed to learn a continuous vector representation of 3D shapes, along with a 'decoder' function, that allows the transformation from this vector space to the original 3D morphological space. We find that approach successfully learns coherent representations: particular directions in latent space are associated with discernible morphological meaning (such as elongation, flattening, etc.). More importantly, learned latent vectors have ecological meaning as shown by their ability to predict the trophic niche of the bird each bill belongs to with a high degree of accuracy. Unlike existing 3D morphometric techniques, this method has very little requirements for human supervised tasks such as landmark placement, increasing it accessibility to labs with fewer labour resources. It has fewer strong assumptions than alternative dimension reduction techniques such as PCA. Once trained, 3D morphology predictions can be made from latent vectors very computationally cheaply. The trained model has been made publicly available and can be used by the community, including for finetuning on new data, representing an early step toward developing shared, reusable AI models for analyzing organismal morphology.

## Author summary

Scientists are now able to gather a wealth of information about the 3D shapes of organisms, which could revolutionize our understanding of how nature and evolution

**Data availability statement:** All data used in the analysis are publicly available from the NHM Data Portal at https://data.nhm.ac.uk/dataset/markmybird. R code for the preprocessing beak meshes can be found at https://github.com/rdinnager/deepMorph/blob/master/R/mesh_to_sdf.R. Python code for running the DeepSDF generative A.I. model can be found at https://github.com/marian42/shapegan/tree/birds. Lastly, R code for running all downstream analysis can be found at https://github.com/rdinnager/deepbills. The model trained in this study is available for reuse or finetuned training at https://github.com/rdinnager/fibre

**Funding:** The author(s) received no specific funding for this work.

**Competing interests:** The authors have declared that no competing interests exist.

influence the forms of living creatures. Yet, to fully unlock this potential, we need new ways to handle and interpret such complex data. In this study, we've employed cutting-edge artificial intelligence (AI) to help sort out and make sense of intricate 3D shape data. To do this, we trained an advanced AI model on a database of 3D scans of bird beaks from over 2,000 different species. The AI was programmed to learn a simplified version of each 3D shape and to understand how to convert back and forth between this simplified form and the full 3D shape. We found that our AI model effectively learned to represent and interpret the forms of bird beaks. It was even able to predict the types of food a bird species might eat based on the simplified representation of its beak. This approach requires less human input and makes fewer assumptions than existing methods, providing a valuable new tool for analyzing animal morphology that complements existing methods and has many potentially promising downstream applications.

## Introduction

Biological data on the phenotypes of organisms can be incredibly diverse and complex, posing significant challenges for statistical analysis and interpretation in organismal biology fields such as ecology and evolution. Despite the seeming complexity, the theoretical principles of evolutionary biology lead us to believe that a set of relatively simple processes (selection, drift, gene flow and mutation) can generate this difficult to wrangle tangle of organismal diversity. One answer to the question of how this happens is development, which allows a relatively simple set of instructions and starting conditions to assemble a complex phenotype through non-linear probabilistic dynamics through developmental time [1,2]. This implies that complex phenotypes could be a high-dimensional realization of an underlying distribution in a lower dimensional, simpler space. Development can be considered a non-linear 'decoder' from this simpler space. The idea that high-dimensional complex data is often a realization of a simpler representation is a significant concept in machine learning, often known as the 'manifold hypothesis' [3]. It hypothesizes that high dimensional data with obvious structure (to us, as humans) actually lies on or near a low dimensional, but potentially highly non-linear structure, known as a manifold. This suggests that there is strong alignment between the computational ideas of manifold learning and ideas in biology about development and the genotype-phenotype map. Regardless of how strong this analogy is in reality, it does imply that the rapidly advancing tools of manifold learning are likely to be very useful for biologists trying to understand the causes and consequences of complex phenotypic data.

Manifold learning has recently made rapid progress through the paradigm of generative AI models, which seek to find an optimal model that can generate a complex high-dimensional data distribution, typically through the use of a lower dimensional latent space representation. Recent work has shown that the latent space of a generative AI model forms a Reimannian manifold that captures a low dimension representation of the high dimensional data [4–7], allowing the methods to capture a simpler form for the data which acknowledges its complexity through its non-linearity (in contrast to simpler methods such as PCA, which assume linearity). These generative AI models have largely been developed in the context of image data, so their potential in biology has yet to be realised, though they are beginning to be taken up within molecular biology for a variety of uses such as visualisation [8], probability density estimation in high dimension [9], generating candidate functional proteins [10] or population genomics distributions [11], and discovering interpretable representations for downstream tasks [12]. To our knowledge, generative AI models have not been applied to complex phenotypic data of organisms.

One such example of complex phenotypic data is 3D shape data. This type of data presents several challenges, which can make analysis difficult and time-consuming.

The challenges of analyzing 3D shape data include:

- High dimensionality: The data can have a large number of variables, making analysis complex.

- Unaligned data across species: Data from different species may not have a consistent structure, making comparisons difficult. Homology is not obvious without some kind of alignment (similar to sequence data when insertion and deletions have happened), or the placment of comparable'landmarks'

- Continuous nature: Theoretically 3D surfaces are continuous but in order to analyse the data it typically has to be discretized in some way, meaning we can have the same theoretical object, but the data can have varying resolution (e.g., number of triangles in a triangular mesh).

Existing methods for analyzing 3D shape data require a significant amount of manual labor and expertise. This includes aligning and choosing landmarks, and even then, data dimension reduction is often required. Commonly used techniques, such as principal component analysis (PCA), can produce data that is not always suitable for downstream analysis, such as phylogenetic comparative methods and statistical models, due to its restrictive assumptions (primarily linearity).

To overcome these challenges, deep generative models hold much promise for extracting meaning from complex biological data. These models have several advantages, including:

- Ability to reconstruct training data: A deep generative model can create a model that can generate the raw data, in this case a full 3D surface, removing the necessity of summarizing the data before analysis. This makes it simple to evaluate the quality of a latent space embedding by visual comparison of reconstructions and the original data.

- Low-dimensional representation: The model can find a low-dimensional representation that conforms to simple distributional assumptions, known as a latent variable space, which is useful for downstream tasks. It has been suggested that generative AI models tend to encode human-interpretable 'concepts' linearly [13], perhaps making them comply better with various downstream analyses' assumptions.

- Embedding new data: New data can be embedded into the model, without retraining.

- Generating or reconstructing'novel' data: New data can be reconstructed from any value of the latent variables, even those that were not observed, through interpolation in the latent space, or sampling from a prior distribution defined over the latent space. This allows easy and fast visualization of trends in the latent space as well as the production of'sythetic' data useful for, as an example, upsampling to reduce data imbalance in downstream analyses.

All the above things are possible to achieve with simpler methods such as Principle Components Analysis, but they often require complicated computational downstream methods, such as methods that 'morph' or bend references meshes into a shape that has PCA scores that match a particular desired values, required expensive optimization algorithms. With generative methods, all the advantages come essentially for free once a model is trained. The model is then reusable for many purposes and can be used to generate data very computationally inexpensively. In this study, we evaluate a deep generative method, specifically the DeepSDF model [14], for analyzing 3D morphology, and apply the results to a downstream task. Our findings demonstrate that latent variables derived from deep generative models of complex

biological data have as much ecological signal in them as simpler methods based on geometric morphometrics, which suggests great potential for future applications in this field given all the above benefits relative to existing methods.

### The 3D morphology of beaks across the avian tree of life

Bird beaks have a stunning diversity of forms that are linked to their feeding ecology [15,16]. At the same time, significant constraints on their form exist due to the requirements of flight (e.g., they must not be heavy) and also due to the existence of strong developmental constraints [17]; [18]. The shape of a bird's beak also has consequences for its ability to deal with climate variations, and for characteristics of its song [19–21], and so having a deep understanding of variations in shape that can capture its full complexity could have very important downstream uses.

We explore the benefits of deep generative models for analyzing 3D morphology using the full 3D shape of the surface of bird beaks, collected from over 2,000 bird species across the avian tree of life.

## Results

### Exploring the latent space

A DeepSDF model was successfully trained on the 3D bird beak mesh dataset, using a single GPU (achieving a mean absolute error of 0.1 over 5000 epochs, note the loss is difficult to interpret given the shape is mostly determined by low error near the surface only). A visualization of the latent space in two dimensions reveals that the model produces a latent space that smoothly transitions between different types of beak shapes (S1 Movie), suggesting that the estimated vectors capture meaningful and continuous variation in beak shape. A comparison of reconstructed beak meshes with original meshes showed good correspondence in overall shape (Fig 1). An examination of all 2,021 reconstructions of the original data suggested that the model captured the overall shape of beaks excellently but struggled to capture finer details such as the nares or elaborate ornaments on some species (see supporting document on figshare with all beak reconstructions: [22]).

### Morphological coherence

To see whether the estimated latent space has morphological'coherence', we discovered vectors through the latent space associated with a bird beak's'elongation' and'broadness'.

Figs 2 and 3 show that the discovered vectors of elongation and broadness have strong and consistent effects. Regardless of which starting beak the elongation vector was applied to, it led to an increase (or a decrease for the negative direction) in how elongated the beak was. The same was true for broadness -- applying the vector led to broader beaks in the positive direction. Moreover, the vectors did not appear to be 'entangled' with other features. Entanglement is the phenomenon, observed in some deep generative models, where some vectors can 'entangle' multiple features together as we move along them. Often features are correlated in the training sample, and so they are in a sense 'unentangled' by the model, which appears to have happened at least for these simple types of features.

### Ecological meaning

To see if the latent vector estimated by the DeepSDF model held ecologically important information, we analysed their association with the trophic niche of bird species' they belonged to.

In Fig 4 we show a plot of the beak morphology latent space, reduced to two dimensions using the UMAP algorithm. Points are coloured by trophic niche which shows, broadly, how trophic niches are distributed in the latent space.

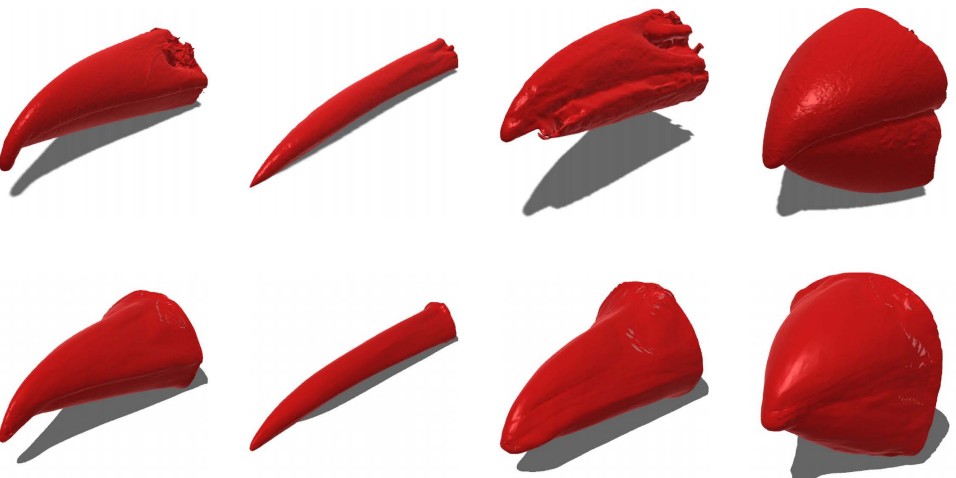

**Fig 1. Some example reconstructions from the DeepSDF model, chosen randomly.** Top row is the original 3D meshes, bottom row is the reconstruction of the same bird bill from the fitted DeepSDF model. The model has captured effectively the overall shape of each bill, even when the mesh has clear defects (the model having filled this in from what it has learned about the full distribution of bill shapes). On the other hand, small details are not captured well such as the nares, but this was not desired in this study.

## Trophic niche prediction

To see whether the latent variables could predict trophic niche we fit classification Random Forest models with trophic niche as the response variable and the 64 latent morphological variables as the predictors.

Training a Random Forest model to predict the trophic niche of a bird based on its estimated latent vector produced a model with an overall 60% accuracy on a test dataset held out from training representing 20% of the original bird beaks. Balanced accuracy, designed to account for imbalanced classes, and defined as the average of specificity and sensitivity, was high (0.78, maximum is 1.0). The discrepancy between the measures is largely explained by the fact that the model did poorly at predicting the trophic niche of omnivores, which has a fairly large number of members (see Discussion). Looking at the confusion matrices of the Random Forests model provides more insight (Fig 5). Most trophic niches had most of their members correctly classified, except omnivores, which had more of their members incorrectly classified, typically as invertivores. Most trophic niches that had incorrect classifications had most of them misclassified as omnivores or invertivores. These misclassifications make sense because invertivores and omnivores appear to have the most widely spread distribution in the morphological latent space (Fig 4).

On the other hand, a Random Forest model trained with 64 PCA axes calculated from the aligned landmark data from Cooney et al. [23] did very slightly worse at classifying trophic niche (60% accuracy, 0.77 balanced accuracy).

We also trained a generalized linear model (with regularization) using the glmnet R package using both our latent variables and PCA scores as predictors. For comparison with the Random Forest models, we used elastic net regularization with the same cross-validation procedure to tune the mixing parameter (alpha) between L1 and L2 regularization, as well as the overall regularization strength (lambda). The multinomial regression model achieved higher accuracy with the DeepSDF latent variables (0.79 balanced accuracy) compared to the PCA variables (0.757 balanced accuracy). This suggests that while the overall predictive power of DeepSDF latent variables is similar to PCA, they may be more amenable to linear classification methods, possibly due to the tendency of deep generative models to encode interpretable concepts linearly in their latent space

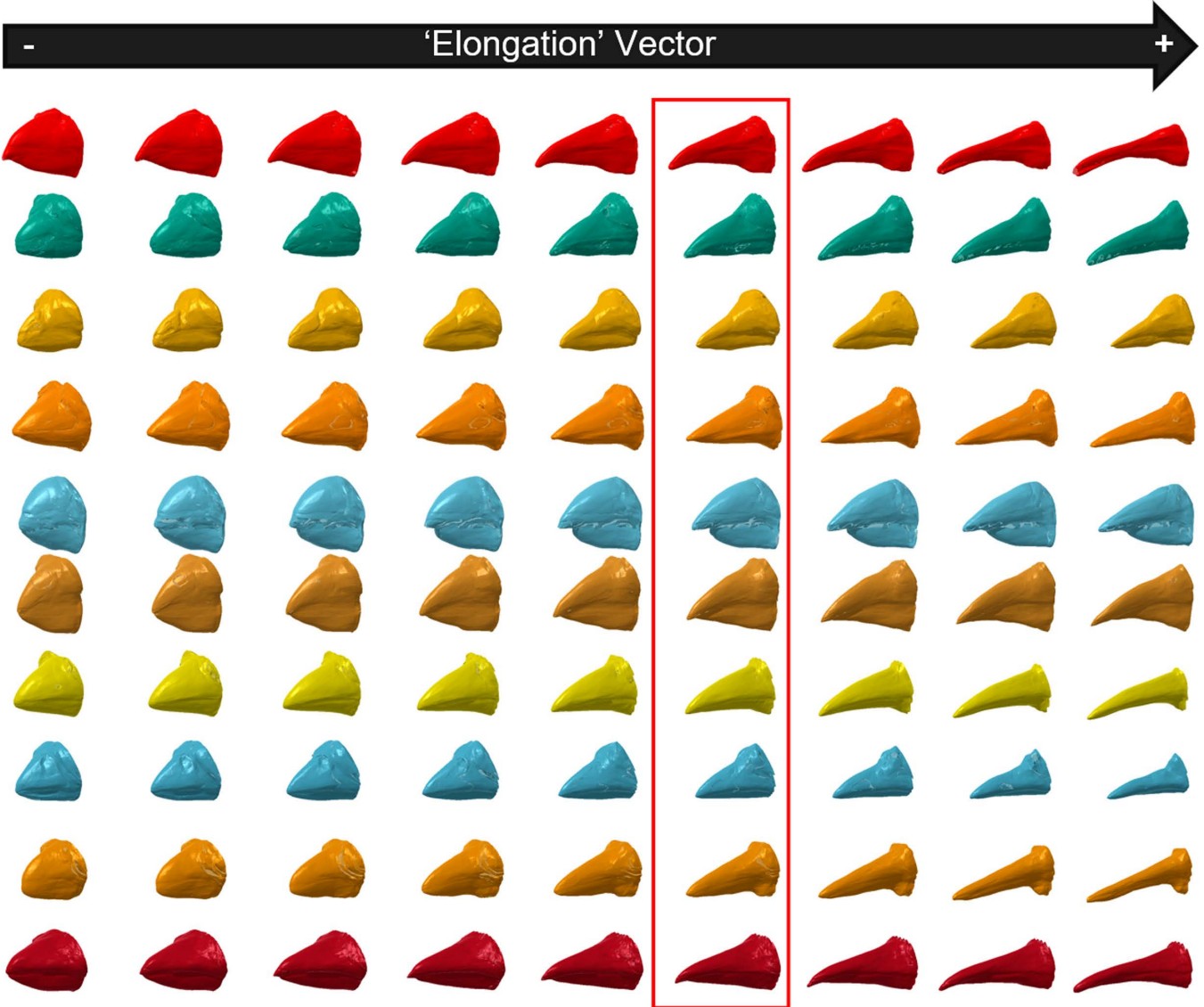

**Fig 2. The latent morphological space contains an 'elongation vector'.** The beaks highlighted in the red box are generated from random latent vectors. The beaks on the right and left are after moving the latent vector along the elongation vector in the positive or negative direction respectively. The elongation vector was found by regressing the latent space against independent measurements of the length, width and depth of the beaks in the observed dataset (see Methods for details).

### Phylogenetic structure and signal

Phylogenetic analysis using the geomorph package [24,25] revealed distinct patterns between PCA and DeepSDF representations. While both showed similar overall phylogenetic signal (multivariate K = 0.23 and 0.16 for PCA and DeepSDF respectively), signal distribution differed markedly. PCA concentrated signal in early axes, and this persisted to some degree even after phylogenetic PCA analysis. In contrast, DeepSDF distributed signal uniformly across all 64 dimensions (Fig 6A).

PACA alignment demonstrated DeepSDF variables could concentrate phylogenetic signal more effectively. The first PACA axis from DeepSDF showed stronger signal (Blomberg's K =

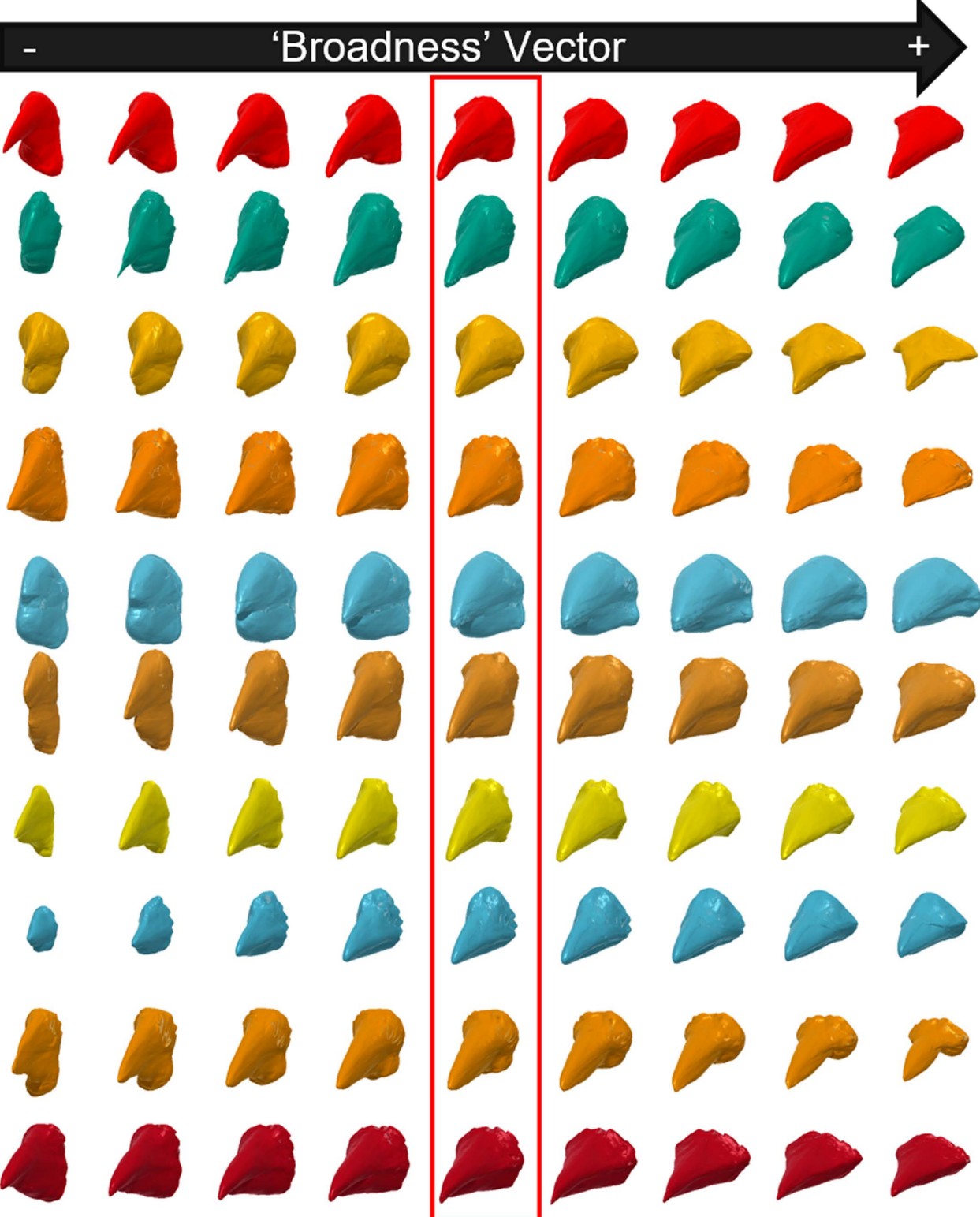

**Fig 3. The latent morphological space contains an 'Broadness vector'.** The beaks highlighted in the red box are generated from random latent vectors. The beaks on the right and left are after moving the latent vector along the broadness vector in the positive or negative direction respectively. The elongation vector was found by regressing the latent space against independent measurements of the length and width of the beaks in the observed dataset (see Methods for details). Note that the highlighted beaks are the same as those highlighted in Fig 2, but viewed from a different angle to make their degree of broadness easier to see.

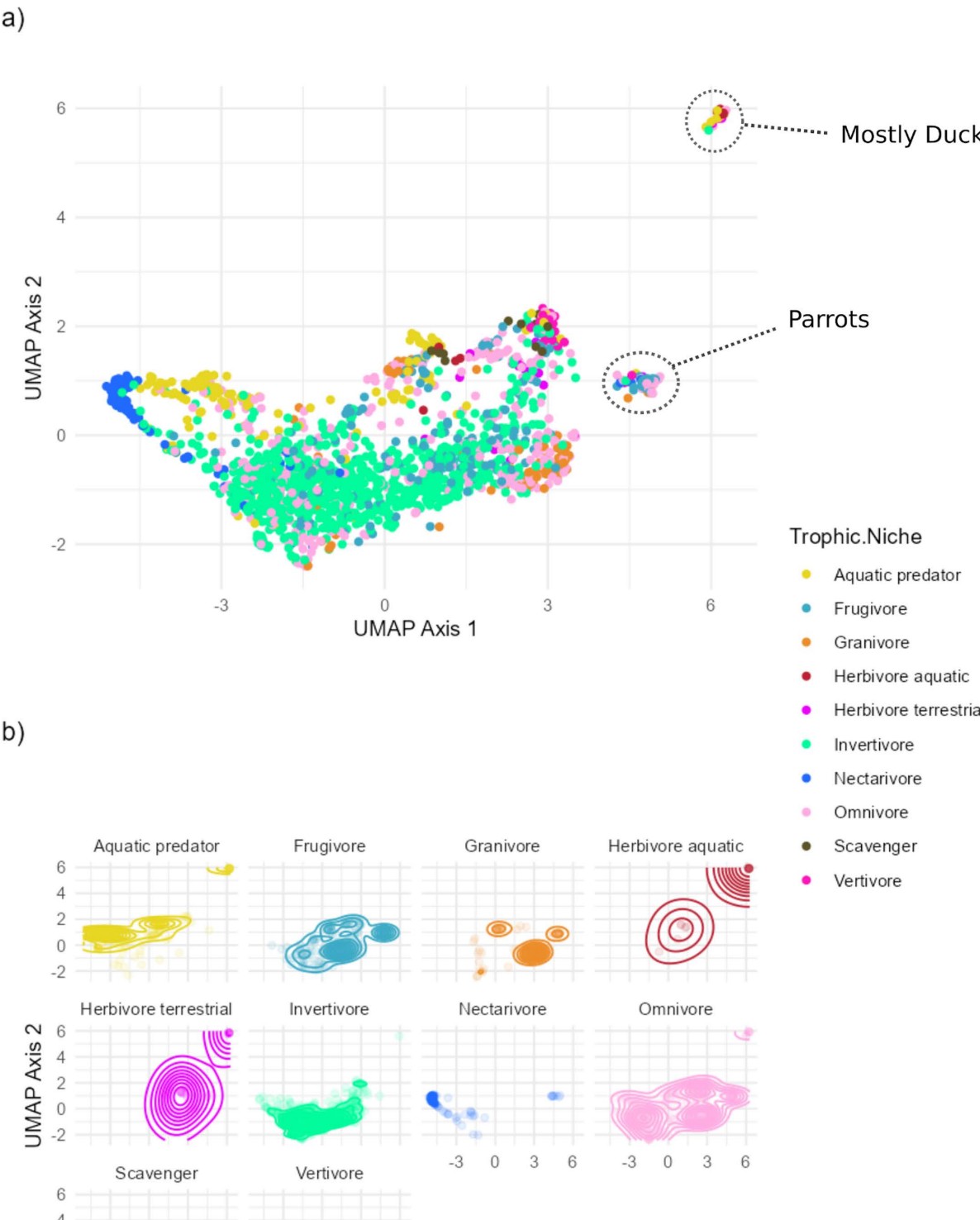

**Fig 4. UMAP dimension reduction of the 64 latent beak morphology variables, plotted by the trophic niche of the bird species each beak belongs to.** We can see good separation for some niches (nectarivores and granivores, aquatic predators). Both invertivores and

omnivores are spread out in their morphology, spanning across most of the space covered by other trophic niches. Panel a shows all bird species, coloured by their trophic niche. Panel b shows density contours for individual trophic niches in the UMAP space. The two most distinct clusters have been labeled with their taxonomic group.

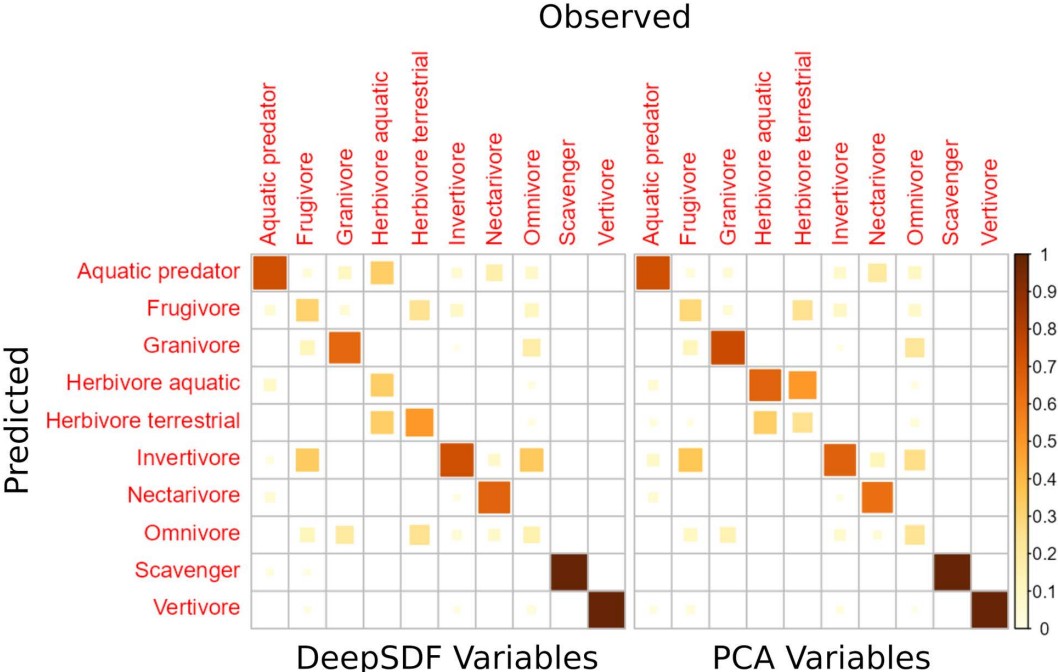

**Fig 5. Confusion matrices of a random forest models trained to predict a bird's trophic niche based on the 64 dimensional latent space estimated by DeepSDF, and 64 PCA variables using landmarks from Cooney et al (2017) [ 23].** Predictions are evaluated on a held-out set of test data that were not used for training the model. On the x axis is the true trophic niches and on the y axis are the predicted trophic niches. Colours and square sizes are proportional to the percentage of observed samples that were correctly classified in each trophic class. The trophic niche that the model predicts the worst is omnivory, which makes sense given we expect this trophic niche to have the most generalized morphology.

0.81) versus PCA (K = 0.57) (Fig 6B). This indicates DeepSDF preserves phylogenetic structure while maintaining uniform distribution across unaligned dimensions.

To see whether phylogenetic signal in latent variables largely explained their ability to predict trophic niche, we repeated the classification analysis using PACA with the first axis removed (hence removing the largest part of the phylogenetic signal). This revealed phylogenetic signal alone did not explain predictive ability for either DeepSDF or PCA. In fact, classification balanced accuracy actually was slightly higher using PACA minus the first axis, and it also increased slightly the advantage of DeepSDF over PCA (balanced accuracy = 0.81 for DeepSDF and 0.78 for PCA). This suggests DeepSDF variables capture additional non-phylogenetic variation relevant to trophic ecology.

## Discussion

### DeepSDF produces a morphologically coherent latent space

The latent space estimated by the DeepSDF model is coherent in that it produces smooth transitions between similar beak morphologies as one moves through it (S1 Movie). Vectors associated with distinct aspects of general beak shape can be found in the latent space (Figs 2

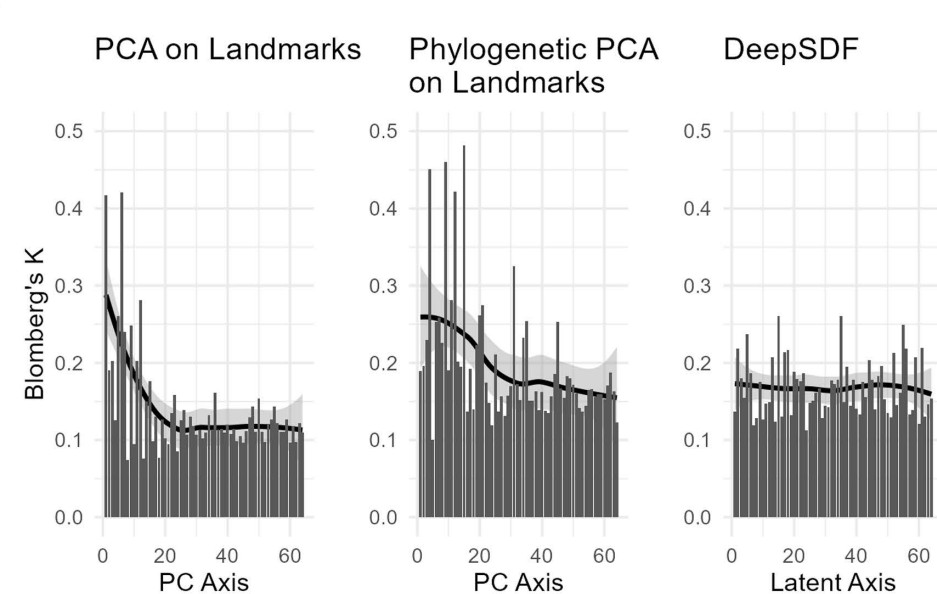

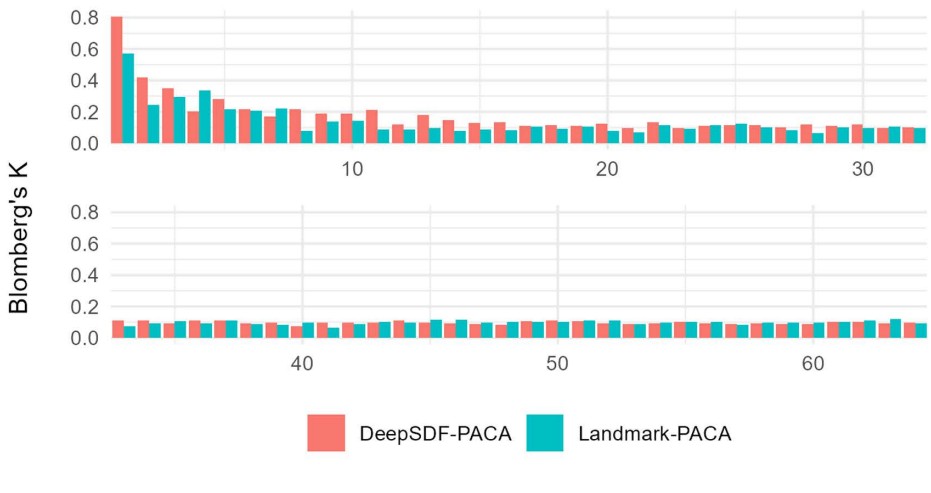

**Fig 6. Phylogenetic Signal Distribution:** a) **Distribution of phylogenetic signal (Blomberg's K) across individual axes for three different representations: standard PCA on aligned landmarks, phylogenetic PCA on landmarks, and DeepSDF latent variables.** Lines show smoothed trends with 95% confidence intervals. DeepSDF shows remarkably uniform distribution of signal across all axes, while both PCA approaches show concentration of signal in early axes despite phylogenetic correction. b) Comparison of phylogenetic signal in PACA components between DeepSDF (coral) and landmark-based PCA (teal). Higher bars indicate stronger phylogenetic signal in that component. DeepSDF components show stronger concentration of phylogenetic signal when explicitly aligned using PACA, suggesting better preservation of phylogenetic structure despite more uniform distribution in the original space.

and [3]), suggesting shape is captured meaningfully by the model. There also is some separation of different trophic niches in a reduced dimensional visualization, though it varies substantially between different trophic niches (Fig 4). We explored this more formally by training a Random Forest model to predict trophic niche from morphological latent vectors.

## Trophic niche is well predicted by morphological latent variables

The results of the Random Forest classification model to predict trophic niche from a bird beaks latent code provided some interesting results. The model correctly predicted the trophic niche of most birds, in both the training set and a held-out test set. Where the model produced incorrect classifications, on the other hand, provides some interesting insight into how bird beak morphology is similar or dissimilar between trophic niches.

For example, the results of the model confirm that nectarivores have highly distinctive beak morphology, since the model almost never incorrectly classified beaks in the nectarivore trophic niche, in either the training or test set. This was also evident in the UMAP dimension reduction (Fig 4), where nectarivores were the most distinct cluster. Our conditional VAE model also produced highly distinctive bird beak samples when asked to generate them while conditioning on the nectarivore trophic niche (Fig 7).

Another interesting trophic niche class was omnivores. Omnivores were frequently misclassified in the test set as invertivores, suggesting that the beak morphology of omnivores is difficult to distinguish from that of invertivores. This is intriguing given that a previous macroevolutionary analysis has shown that omnivory appears to be an evolutionary 'sink' [26]. Burin et al. (2016) showed that omnivores frequently evolved from most other trophic niches, except for invertivores. Once they had evolved, omnivores had a low diversification rate and had almost no transitions into other trophic niches. However, the one exception was invertivores -- omnivores occasionally transitioned to invertivores. The same study also showed that most omnivores had some invertebrates in their diet. Taken together, this suggests that omnivores are in an evolutionary transitional state from other trophic niches to invertivory, but they rarely complete this transition in the macroevolutionary record. Given that omnivores are frequently misclassified as invertivores in the Random Forest model, this could suggest that the transition to invertivory in beak morphology is, in fact, almost complete in most lineages of omnivores. This suggests a speculative explanation for omnivory appearing as an evolutionary 'dead-end'. It could be that the final transition to invertivory from omnivory might be prevented by heavy competition with incumbent invertivores during the final evolutionary approach, leading to frequent competition-driven extinction. Though beyond the scope of this paper, there may be ways to test this speculative theory using a combination of macroevolutionary transition models and the latent beak morphology data provided by this study, perhaps by taking advantage of a conditional or joint VAE trained on the DeepSDF vectors, similar to the one we used for generation in this study. It is also interesting to note that our conditional VAE model generated highly diverse beak shapes when conditioning on the omnivore trophic niche, supporting the idea that omnivores have been derived from a diversity of different ancestral trophic niches (Fig 7).

On the other hand, the DeepSDF latent variables did not perform considerably better than simpler measures of morphology provided by Principle Components Analysis. This means we can say only that the DeepSDF latent variables contain as much predictive power for trophic niches as a simpler linear method like PCA. Furthermore, our results are quite consistent with those of [27], who used PCA scores from only 4 measurements of beaks to predict trophic niches of birds. Like us, they found that omnivores and frugivores were most difficult to predict. However, they discovered that when they also used measurements of leg morphology as an additional set of predictors, that the model was able to distinguish frugivores and a number of smaller trophic classes better than before. This suggests that trophic niche prediction is not an especially difficult task, with the exception of predicting the omnivore class, meaning that it perhaps is not surprising that it was unable to distinguish between DeepSDF and the simpler PCA based method. However, it seems unlikely that any better representation of beak morphology, no matter how complex, will improve trophic class prediction since omnivore

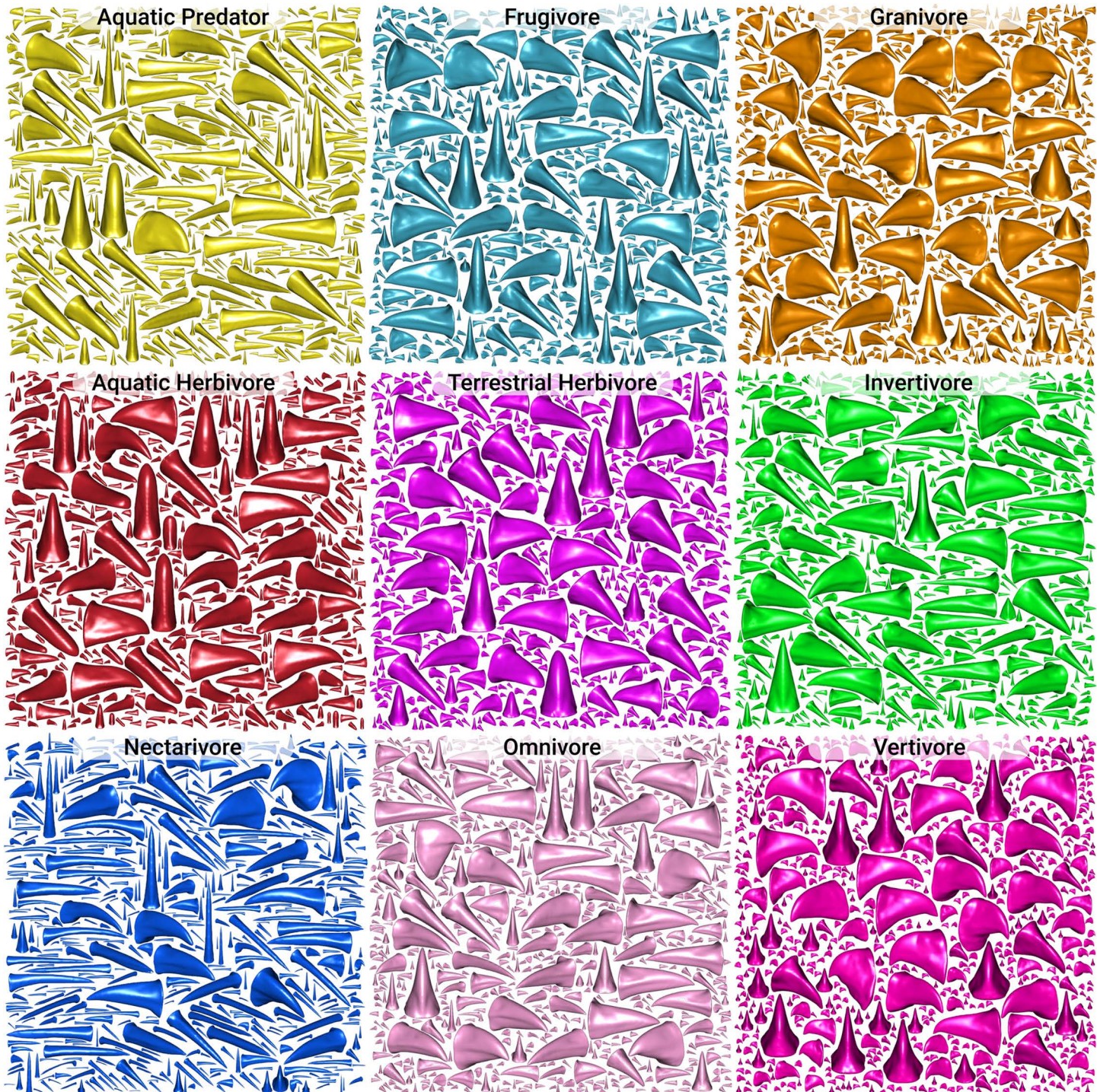

**Fig 7. Probablistic generative samples from a conditional variational autoencoder trained on the latent space of the bird beak DeepSDF model, and conditioned on the trophic niche of the bird species each beak belonged to.** Each panel represents a random sample from the estimated conditional probability distribution given a particular trophic niche. The trophic niches with the most distinctive generated distributions tend to correspond to the trophic niches that have the lowest false negative rate for the Random Forest classifier (Fig 5). For example, Granivores, Nectarivores, and Vertivores stand out with distinctive generative distributions and had low false negatives in the classifier. On the other hand, generative distribution with a lot of beak shape diversity correspond to trophic niches that had low classification accuracy (e.g., Omnivores, Invertivores, Frugivores). Images were generated using the R package impac (https://github.com/rdinnager/impac).

are probably not a good natural class, that is, it is transitional between other classes with more extreme diets. Therefore we expect it to be highly heterogeneous in the morphology of its members and it may not ever be possible to predict it well. What is needed is more fine-grained data on bird diets that includes relative percentages of different diet items. Predicting this sort of data would provide a much more informative test of different numerical representations for beak morphology.

An intriguing result was that when we used a simpler linear model for the trophic classification task, the DeepSDF variables performed better than the PCA scores. This could be a result of a general phenomenon that has been observed in deep generative models, that their estimated latent spaces tend to separate human-interpretable 'concepts' *linearly*. This is sometimes referred to as the 'linear representation hypothesis', which has evidence from deep generative models in many domains, such as language models, and machine vision models. Ref. [13] provides and interesting discussion of linearity of representations along with many references to papers that discuss the phenomenon in various data domains. The consequences of this for any downstream tasks are more difficult to reason about. It may be that this may help the latent representations satisfy the assumption of certain downstream analyses better (such as linearity assumptions). The fact that the representations are by construction Gaussian in distribution also contributes to this possible benefit.

## Properties of the latent space

Like many modern dimension reduction techniques, the DeepSDF method can produce different latent space configurations across training runs while preserving the underlying relationships between specimens. This property is shared by other widely-used methods in biological data analysis, such as non-metric Multidimensional Scaling (NMDS), t-SNE, and UMAP. While methods like Principal Components Analysis (PCA) produce deterministic results, they achieve this by making strong assumptions about linearity and orthogonality that may not reflect biological reality. The non-deterministic nature of these more flexible methods is a reasonable trade-off for their ability to capture complex non-linear relationships in biological data. What matters is not the exact coordinates, but that the method reliably captures meaningful biological variation and relationships between specimens.

If a canonical representation is desired there are a number of options that involve rotating the latent space to a specific repeatable orientation. An example might be to rotate the latent space to align with the variation implied by phylogeny of the organisms under consideration, if available (see following section). Or it could be rotated to align linearly with any chosen set of summary variables, such as beak elongation, or trophic niche. All rotations are equally valid, but such a rotation could facilitate comparison between different models. It is important to note however, that vectors in the latent space are what is meaningful in this approach and this likely better reflects the biological reality, since evolution does not 'see', or act, on variation aligned to our preconceived notions of interpretability (nor along axes of maximum variation, which is what PCA ultimately does).

## Phylogenetic signal and latent space structure

A key challenge in analyzing morphological data is handling phylogenetic signal in multivariate analyses. Researchers often use phylogenetic PCA not to remove phylogenetic signal (which is impossible), but to redistribute it more evenly across axes rather than having it concentrated in the first few dimensions [25]. This redistribution is important because many downstream analyses use only the first few PC axes to represent morphological variation - if these axes concentrate phylogenetic signal, it can violate assumptions of subsequent comparative analyses.

Our analyses revealed an intriguing property of the DeepSDF latent space: it naturally distributes phylogenetic signal almost perfectly uniformly across all 64 dimensions, in contrast to standard PCA which concentrates signal in early axes (Fig 6). Even after phylogenetic PCA correction, some concentration of signal remains in PCA. The uniform distribution in DeepSDF is likely due to the spherical prior used in training, which preferences no directions in latent space over any other. Remarkably, this means DeepSDF achieves automatically what phylogenetic PCA attempts to do through explicit correction - preventing phylogenetic structure from being concentrated in dominant axes.

When explicitly aligning variables with phylogeny using PACA, we found that DeepSDF variables could concentrate phylogenetic signal more effectively than PCA variables. The first PACA axis from DeepSDF showed stronger phylogenetic signal (Blomberg's K = 0.81) compared to PCA (K = 0.57). This suggests that while DeepSDF naturally distributes phylogenetic signal evenly, it preserves the underlying phylogenetic structure in a way that allows it to be concentrated along specific directions through rotation when desired. As Collyer and Adams note [25], this flexibility in representing phylogenetic structure - being able to either distribute it evenly or concentrate it through rotation - is valuable for downstream comparative analyses.

The ability of DeepSDF to maintain predictive power for trophic niches even with the first phylogenetically aligned component removed (balanced accuracy = 0.81 vs 0.78 for PCA) suggests it also captures meaningful non-phylogenetic variation in beak shape. This addresses a key concern raised by reviewers about whether the latent space merely reflects phylogeny. Instead, we find that DeepSDF learns a representation that naturally separates phylogenetic structure from other sources of variation, while preserving both. This property makes it particularly suitable for comparative analyses where researchers need to consider both phylogenetic and non-phylogenetic patterns.

This balance of even distribution and preservable structure aligns with recent work showing that deep generative models tend to encode interpretable concepts linearly in their latent spaces [13], as also discussed above. The spherical prior encourages even distribution of variation across dimensions, while the continuous nature of the decoder allows preservation of phylogenetic structure that can be recovered through rotation. These properties emerged naturally from the training process rather than being explicitly enforced, suggesting DeepSDF has learned a biologically meaningful representation of morphological variation.

## Why use a generative model?

We have established that the representations learned by a deep generative model are at least as good as previous methods, at least for one ecologically meaningful task, suggesting they contain ecological 'signal'. But if a method such as the seemingly much simpler PCA can do as or almost as good, why should we consider using generative models instead? Generative models have a number of advantages from both a conceptual and a methodological perspective. From a conceptual perspective, as we discussed in the Introduction, there is a elegance to the idea of utilizing a model that attempts to learn a generative process for the data. In some sense this mirrors how we think the real biological features that our data measures have come into being. It also allows us to process the raw data much less before we analyze it, such that outputs of our analysis remain 'closer' to the real biological objects of interest. That is, despite the potential complexity of the underlying generative model we are fitting, there are fewer layers of abstraction we have to go through to get our data into an analyzable form.

Methodologically, a major advantage is the ability to quickly and easily generate data from the model in the original data mode, for visual validation or other purposes, such as synthetic data creation.

Another substantial advantage is the reusability of the model. That is, the model as trained can be taken and fine-tuned using new datasets. The depth of fine-tuning can also be manipulated. It is possible for example to keep the decoder function fixed and optimize latent codes for new data into the existing latent space. On the other hand, it is also possible to fine-tune the decoder neural network weights as well, allowing new data to update the latent space itself. While not yet a true foundation model, this work demonstrates the potential for developing shared, reusable AI models in organismal biology. Foundation models typically require massive, diverse datasets and extensive validation across multiple tasks [28,29]. Our current implementation, trained on bird beaks alone, represents an initial exploration of how such approaches might eventually be developed for morphological analysis. The model is publicly available and can be fine-tuned on new data, allowing us to explore the challenges and opportunities of building shared morphological models. As the diversity and scale of 3D morphological datasets grows, and methods for aligning different types of morphological structures improve, this type of approach could potentially evolve toward true foundation models. We can see the transformative potential of such models in molecular biology, where tools like AlphaFold [30] have revolutionized protein structure prediction. This approach is in stark contrast to most existing types of models in organismal biology, which are trained or estimated by individual researchers and are not generally reusable. Unlike our generative AI model, incorporating new data into these traditional models typically requires retraining the model from scratch, making it much more difficult to accumulate improvements over time. However, significant work remains to develop appropriate architectures, training approaches, and validation methods for AI-based morphological foundation models.

Several expert reviewers of this paper brought up the issue of applying this generative AI method to smaller or more constrained datasets, as they were concerned that the method would require large general datasets to be useful. The answer to this is two-fold. For one, though we trained the model presented here on over 2000 bird beaks, we believe the model will still perform adequately on much smaller datasets of upwards of several hundred 3D meshes. To some extent, this will depend on how heterogeneous the datasets are. Datasets with fairly constrained variation will likely require fewer samples to capture the main patterns in the data. On the other hand, small datasets can be accommodated, as discussed above, by using the fine-tuning approach. Now that this model has been trained on a large dataset, it can be reused through fine-tuning on smaller datasets. This is one reason we have made the pretrained model available through the R package 'fibre'. Anyone can take our model, and a set of new bird beaks meshes, and fine-tune the model on the new dataset, taking advantage of what the model has already learned from the larger dataset. This will not just embed the new data into the latent space already learned, but also adjust the underlying model to take into account the new information from the new data, thus further improving performance on any downstream tasks. In fact the benefits of using pretrained models to improve performance on small datasets for various downstream tasks such as classification or regression is well-known and widely utilized within machine learning, and is often referred to as 'semi-supervised learning' [31] or 'transfer learning'. Importantly this does not require access to the original bird beak meshes, which is an enormous and unweildy dataset that takes a fair time to download (in contrast our pretrained model is stored in a file just over 4mb). Of course, this will require care that any new bird beak 3D mesh data is preprocessed in the same manner that we processed our data. This can be achieved by modifying the code we provided publicly. The first author of this paper can also be contacted for help in this task. It is likely the model presented here may need to be further developed before it can become a truly useful foundation model, but we hope that what we present here is a useful starting point.

## Future work

The method shown here represents a promising new approach for analyzing 3D morphological data, though significant development work remains. There are many downstream tasks that could be enabled by the flexibility of the model, as well as many ways the model could be improved or augmented. Here we talk about some of these possibilities.

**Improving the model.**  In the time since the DeepSDF model was first introduced by Park et al. (2019), methods for modelling 3D objects using signed distance functions has advanced at a fast pace. Already several more advanced models exist that improve the ability of the model to capture very complex and intricate shapes, including whole scenes. This could allow the method to be applied to extremely complex morphologies of organisms, such as the inner ear or even part of the body with articulated joints. For example, geometric implicit representation models have been show to represent very complex shapes by modelling the SDF implicitly. That is, it estimates the function by optimizing a function that is zero near the surface of the object but which has a gradient in 3D space that aligns with the surface normals of the object and which is nearly one everywhere [32].

In this study we found that the overall shape of the beak was well captured (the primary goal of this study), however finer details such as the nares were not well-recovered (Fig 1). If this detail was desired, this could probably be achieved in several potential ways: 1) Increasing the size of the latent space; 2) increasing the complexity and thus capacity of the encoder and decoder networks, or 3) use a newer method such as the geometric implicit representation method described above, which has been shown to be able to capture finer 3D detail.

**Phylogenetic ancestral predictions.**  Because a 3D shape can be reconstructed from any vector in the model's latent space, latent vectors associated with extant species might be ideal for reconstructing beak shapes of bird common ancestors. This could be accomplished by treating estimated morphological latent variables as hypothetical or latent 'traits', which could then be modeled using standard phylogenetic methods for estimating ancestral states of continuous variables. Once ancestral states of the latent vector have been estimated, these could be plugged into the DeepSDF neural network we've already trained to get a prediction of the full 3D shape of that ancestor. Though beyond the scope of this paper, exploring this idea will be the subject of a follow-up study.

In addition to using estimated vectors in a downstream phylogenetic analysis, an alternative and potentially powerful approach could be to incorporate a phylogenetic model directly into the DeepSDF model. This could potentially be accomplished by estimating a set of evolutionary change vectors along each branch of a phylogeny linking the species to be modelled, within the model. The final latent vectors are then constructed by summing these vectors along each edge of the phylogeny from the root to the tips, and these tip vectors are then fed into the decoder neural network instead. The prior distribution is then applied to the trajectories, instead of the tip vectors. With a simple Gaussian as the prior on independent trajectory components we have a model equivalent to a classic multidimensional Brownian motion model, in the latent space. This will be the subject of a future study.

**Adding new data.**  New data can be embedded into the latent space of an already trained model. This is achieved by generating an SDF sample from a new 3D mesh that can be compared to reconstructions from the latent space. It works by starting with a random vector in the latent space, feeding this to the decoder network to generate a predicted SDF at the same coordinates of the SDF sample we want to 'find' in the latent space. We can then calculate how close the SDF prediction are to the observed SDF of interest using mean squared error. From this we can calculate a gradient back through the decoder model to the latent variables, which tells us which direction to move the latent vector to improve the fit. We can then use standard gradient based optimization to search the latent space for the best fit to the

new data, allowing the new bird beak to be embedded in the existing latent space. It would be interesting to see where bird beaks not in the training set will fall in the latent space, and how well the latent space represents these new beaks. That is, it could serve as a test of the ability of the learned latent space to generalise to unseen bird beak shapes.

Because we wanted to use all available data to estimate this model, we did not have a held-out test set to look at this question. However, the Mark my Bird project which provided the 2020 bird beak scans used in this study has continued its work since releasing that data. Eventually the project will release scans for the remaining 7000 or so birds that have been scanned to date. This will be an ideal opportunity to see how this sample generalizes to a much larger number of species.

Another intriguing possibility would be to see where extinct bird species might fall in the latent space. This is challenging because most extinct bird beaks are only know from fossilized remains, which only include the bones in the beak, whereas the scans we work with here include the membrane covering the beak found in live preserved birds. However, museum samples for species that have gone extinct recently do exist, such as for the American passenger pigeon, which could in theory easily be scanned and included.

**Discovering more meaningful latent vectors.**  Using a similar approach to how we discovered the elongation and broadness vectors, we could look for association in latent space with other factors of interest in birds, as long as data is available. For example, we could create a dataset where we score a sample of bird beaks with its 'degree of hookedness', and use this to see if a vector in the latent space is associated with how much a beak is hooked. There are also 'unsupervised' methods that have been developed for finding meaningful latent directions in deep generative models (e.g., [33]), and it might be useful to develop similar methods for this approach.

## Materials and methods

### The dataset

We used a set of 3D scans of museum specimen birds that was described in [23]. The data comes in the form of 3D meshes, produced by scanning the front of the head of bird specimens using a 3D scanner, and is publicly available at www.markmybird.org.

**Mesh preprocessing.**  The raw 3D mesh data was cleaned and preprocessed with an algorithm written in R (utilizing the R packages rgl [34], Rvcg, Morpho [35], and proxy [36]) that conducted the following steps:

1. Preprocessing: The input mesh is cleaned and optimized to ensure that it is manifold and free of self-intersections or degenerate faces.

2. Reorientation: The mesh is reoriented so that the beak region is aligned with a predefined reference frame, ensuring consistency across different input meshes. The beaks were aligned so that a line drawn from the tip of the bill to the center of the bill's base, where it connects to the head, were all aligned across all bills. Additionally a line running from the bottom center of the bill's base to the top center was also aligned across all bills. These lines were derived by connecting three landmarks that were placed by volunteers and which were also made publicly available. Though we used landmarks for orientation because they were available, alternative, landmark-free methods exist for aligning 3D meshes (e.g., [37]), which could be used to reduce the requirement for placing landmarks to zero, since landmarks are not required for any subsequent part of this analysis. In any case, we only used three of the dozens of landmarks created by volunteers, representing far less total labour.

3. Trimming and Hole filling: The meshes included part of the bird's head where the bill attached to the head. Additionally the back of the meshes that faced away from the 3D

scanner were open. Since the method described below requires watertight meshes, we trimmed the head away from the bill and closed the resulting hole. To do this we generated a sphere that perfectly enclosed a set of points on the convex hull of the bird's bill (the bill tip, the upper, lower, left, and right base of the bill). We trimmed any mesh that fell outside this sphere and then knit the the surface of the sphere to the bird bill where the hole at the back was, creating a standardized curved surface at the back of each bill.

4. Post-processing: The resulting trimmed mesh is post-processed to repair any topological inconsistencies introduced during the trimming process, such as non-manifold edges or disconnected components. This ensures a high-quality output suitable for further analysis or visualization.

## The signed distance function

The signed distance function of a 3 dimensional shape is a unique function that allows the reconstruction of the shape from 3D coordinates. The function takes a three dimensional vector, $x$, representing a coordinate in 3D space as input, and it outputs the signed distance between that coordinate and the closest point on the surface of the 3D shape, $s$. We refer to this function as:

$$\mathrm{SDF}(x) = s : x \in \mathbb{R}^3, \ s \in \mathbb{R} \tag{1}$$

The sign of the output represents whether the coordinate is inside or outside the shape (negative being inside, and positive outside). The full 3D surface of the shape is then implicitly represented by the zero isosurface of the function, that is, the infinitesimal region in which the function output goes from negative to positive (or from inside to outside the shape, $\mathrm{SDF}(.) = 0$). Park et al. (2019) present a method to estimate such a function using a deep neural network to approximate the function itself, and a Monte Carlo sample of the signed distance field of the 3D shape(s) to serve as data for the optimization of the function parameters. The function can be augmented with a set of learned latent vectors that represent each of a set of individual 3D shapes that can be trained simultaneously. A coherent structure of the latent space is encouraged with a simple prior distribution that maintains compactness and sphericity.

In practice, the signed distance function for the 3D surface of bird bill $i$ ($\mathrm{SDF}_i$) is approximated by a function with parameters $\theta$:

$$f_\theta(z_i, x) \approx \mathrm{SDF}_i(x) \tag{2}$$

where $z_i$ is an $n$-dimensional vector that will be optimized to represent the unique shape of bird bill $i$.

The parameters of the function are optimized by comparing its output to a dataset composed of a spatial sample of $n_S$ 3D coordinates per bird bill, paired with their true SDF value, sampled over all target bird bill shapes indexed by $i$ ($S$):

$$(x_{ij}, s_{ij}) : \mathrm{SDF}_i(x_{ij}) = s_{ij} \ \forall 1 \leq i \leq n_N, 1 \leq j \leq n_S \tag{3}$$

where $n_N$ is the total number of bird species and $n_S$ is the total number of coordinates sampled per bird species.

## Model specifics

The goal of the model is to approximate the following posterior distribution:

$$p(z \mid s, x) \propto p(z)\, p(s \mid x, z) \tag{4}$$

Assuming that deviations of the estimated SDF from the true SDF follow a Gaussian distribution, and that the latent vectors $z$ have a prior probability that is multivariate normal distribution with diagonal covariance matrix (e.g., spherical) we can express the model as:

$$s_{ij} \sim N\!\left(f_\theta\!\left(z_i, x_{i,j}\right), \sigma^2\right) z_i \sim N\!\left(0, \phi^2 I\right) \tag{5}$$

In general the above posterior distribution is intractable but can be approximated efficiently in several ways. In deep learning applications a variation approximation can be used to estimate the full posterior (such as in, e.g., variational autoencoders). Here, however, training is simplified by only estimating the maximum a posteriori (MAP) of the distribution. In the following we assume $f_\theta(z, x)$ accepts a matrix input z with $n$ columns of rowwise concatenated vectors $z_i$, and likewise $x$ is a matrix with the 3 columns and the same number of rows as z, and it outputs a column vector of approximate SDF values with a length equal to the number of rows of z and $x$. $s$ is a column vector of true SDF values. Given this we have:

$$p(zs, x) \propto p(z) \cdot p(sx, z)$$

$$\propto \exp\!\left[-\frac{1}{2}(z - 0)^T \frac{1}{\phi^2} I (z - 0)\right] \cdot \exp\!\left[-\frac{1}{2}(s - f_\theta(z, x))^T \frac{1}{\sigma^2}\left(s - f_\theta(z, x)\right)\right] \tag{6}$$

$$= \exp\!\left[-\frac{1}{2\sigma^2}(s - f_\theta(z, x))^T \left(s - f_\theta(z, x)\right) - \frac{1}{2\phi^2}\| z \|_2^2\right]$$

Maximizing $z$ gives:

$$\operatorname*{argmax}_{z}\quad \exp\!\left[-\frac{1}{2\sigma^2}(s - f_\theta(z, x))^T \left(s - f_\theta(z, x)\right) - \frac{1}{2\phi^2}\| z \|_2^2\right]$$

$$= \operatorname*{argmin}_{z}\quad \frac{1}{\sigma^2}(s - f_\theta(z, x))^T \left(s - f_\theta(z, x)\right) + \frac{1}{\phi^2}\| z \|_2^2 \tag{7}$$

$$= \operatorname*{argmin}_{z}\quad \sum(s - f_\theta(z, x))^2 + \frac{\sigma^2}{\phi^2}\| z \|_2^2$$

Therefore we maximize the MAP by mimimizing the penalized likelihood expression

$$Q(z \mid s, x) = \mathcal{L}(s \mid z, x) + \frac{1}{\phi^2}\| z \|_2^2 \quad \mathcal{L}(s \mid z, x) = \sum(s - f_\theta(z, x))^2 \tag{8}$$

$L$ is the standard sum of squared errors between the true SDF values and those predicted by the function $f_\theta$, known as the reconstruction loss. The penalty term is just the squared deviations of $z$ from zero (across all $n$ dimensions), multiplied by a parameter $\lambda = \frac{1}{\phi^2}$, very similar to the L2 loss term over coefficients in a ridge regression. $\lambda$ is a hyper-parameter of the model that can be changed to adjust the relative influence of the reconstruction and the prior to the loss.

In practice it is desirable to make sure the model correctly estimates the SDF near the surface of the 3D shape, since it is only the zero isosurface that ultimately determines the

shape, and so finding very accurately where the SDF turns from positive to negative is the most important task. Therefore, in practice, the model is trained to focus on reconstructing the SDF close to the surface by clamping the observed and predicted SDF within a small range around zero, which is controlled by a hyper-parameter $\delta$ (which was set to 0.1 in this study). Therefore, instead of using the Gaussian $L$ function, we use the following reconstruction loss instead:

$$L(sz,x) = \sum \left( \left| \text{clamp}(s) - \text{clamp}\left(f_\theta(z,x)\right) \right| \right) \tag{9}$$

where $\text{clamp}(x,\delta) := \min\left(\delta, \max(-\delta, x)\right)$. Note this is possible because the formulation of $Q$ derived above is not dependent on the Gaussian form and holds for any factorizable reconstruction loss function, as shown in Park et al. (2019). This penalized likelihood can be efficiently optimized, jointly over the $z$ vectors and the parameters $\theta$ of the function $f_\theta$ using the stochastic gradient descent algorithm.

## Autodecoder architecture

Note that using a fixed $z$ based on MAP estimation, instead of a full posterior fit using a variational approximation, was shown to improve training results by Park et al. (2019). The overall structure of this model is what is known as an 'autodecoder' in the machine learning literature. This distinguishes it from autoencoder and variational autoencoder architectures, which are more commonly used in generative AI modelling. Park et al. (2019) suggest a major advantage of a decoder-only model architecture is that $z$ can be estimated using partial batches of independent SDF samples from any number of 3D meshes simultaneously, whereas models with autoencoders must process a set of an entire 3D meshes at once, because the encoder part of the model needs to capture the global structure of each of the meshes. The disadvantages of the decoder-only architecture is that we lose the benefit of amortized inference for large datasets, and we lose the ability to model uncertainty in the conditional probability distributions (and as a side-effect lose the power of variational autoencoders to find a minimal low-dimensional manifold to explain the data). The first issue is not a problem in our case because the full dataset of ~2,000 bird beaks is relatively small for a machine learning dataset and so amortized inference is not necessary. We mitigate the second problem by using a second-stage conditional variational autoencoder (VAE) model that we fit on the estimated $z$ vectors, which is inspired by recent work on 'two-stage VAEs' [38,39], which allows us to produce a full probabilitic generative model (see Trophic Niche Conditional VAE section for details).

**Neural network architecture.** The function to approximate the SDF, $f_\theta$, is a trained neural network. We use a slightly modified version of the architecture described in Park et al. (2019), which is described briefly here. We use two chained multilayer perceptron modules with 3 layers each as the primary network architecture. The input to the first module is a vector of 3 coordinates in 3D space concatenated with a latent vector of length 64, representing the bird species whose beak the current coordinate belongs to. Each hidden layer in the first module has 512 neurons. The input to second module is the output of the first module concatenated with the 3 coordinates and the length 64 latent code vector (a skip connection). This helps the network 'remember' the original information as it goes through the neural network layers, and was found to improve performance in Park et al. (2019).

**DeepSDF model training.** The model was trained by minimizing the penalized likelihood loss function described above, using minibatches of SDF samples generated from preprocessed bird beak meshes. Samples were generated in R by sampling points uniformly from a unit sphere that enclosed the bird beak, as well as an equal number concentrated near

the surface of the beak, in order to encourage the optimization to focus on this area, given it is the most important for reconstructing the zero isosurface. Points close to the beak surface were generated by sampling uniformly over the surface (using the Rvcg package) and then adding some small Gaussian noise to spread them out from the surface. Sampled points were saved to disk and loaded by the Python script used for fitting the model. The loss function was optimized using stochastic gradient descent, as implemented in the Pytorch package of Python [40]. Training was conducted on a single Nvidia A100 GPU, and was continued on new SDF samples in each epoch until the reduction in loss from epoch to epoch no longer showed improvement.

**3D morphology reconstruction.** Once the model is trained, 3D surfaces can be reconstructed by spatial sampling of coordinates, SDF prediction of the coordinates, and an algorithm to detect where the zero isosurface lies. More specifically, we reconstruct a 3D mesh using regular spatial sampling to generate a set of voxels predicted to be inside the surface, which is then converted to a mesh using the Marching Cubes algorithm [41]. A high resolution rendering of the shape can by achieved using the Raymarching algorithm, which uses the SDF function (and its gradient) directly. Both algorithms were implemented in R programming language (using the torch package and the rmarchingcubes package).

**Implementation.** The DeepSDF method was implemented in PyTorch for Python [40], and the code is available at https://github.com/marian42/shapegan, in the 'birds' branch of the github repository. A description and exploration of the implementation, as well as an exploration of some alternative generative models for 3D shapes can be found in [42]. Additionally, the model was reimplemented in R, using the 'torch' package [43]. The model trained in Python was then imported into R and is available in the in-development R package 'fibre' (https://github.com/rdinnager/fibre). The R pretrained model can be used to generate 3D beak meshes for any inputted latent vector, and was used for all downstream analysis reported in this study, which were conducted using R. The model will eventually be made available in a new R package that will also provide a user-friendly set of functions to allow training of a similar model on user's own datasets, or finetuning the existing model.

## Trophic niche conditional VAE

In order to create a fully probabilistic generative model for bird beaks, we fit a second-stage variational autoencoder, partially based on the ideas of 'two-stage VAEs' [38,39]. Essentially we used the estimated latent vector from the DeepSDF model as input to a conditional variational autoencoder (VAE).

VAE models in general are a highly popular and powerful method in generative A.I. that we very briefly describe here. Variational Autoencoders (VAEs) are a class of deep generative models which seek to capture the underlying probabilistic distribution of complex, high-dimensional datasets [44]. VAEs consist of two main components: an encoder network, which maps the input data to a latent lower-dimensional space, and a decoder network, which reconstructs the original data from this latent representation. The encoder network approximates the posterior distribution of the latent variables, given the input data. This is typically modeled as a multivariate Gaussian distribution, parameterized by a mean vector and a covariance matrix which are outputs of the encoder network. The decoder network then generates data by sampling points from this distribution and mapping them back into the original high-dimensional space. The aim is to learn a set of latent variables that can accurately and efficiently represent the input data.

The key feature of VAEs is their use of a variational loss function during training, the evidence lower bound (ELBO) on the log-likelihood of the data:

$$\log p_\theta\left(x\right) \geq \mathbb{E}_{q_\phi(z|x)}[\log p_\theta(x\,|\,z)] - D_{KL}(q_\phi(z|x)\|p_\theta(z)) \tag{10}$$

which is a sum of two terms: a reconstruction loss and the regularization term, also known as the Kullback-Leibler (KL) divergence. The reconstruction loss quantifies how well the decoder is able to reconstruct the original data from the latent representation. The KL divergence term acts as a regularization component, which measures the divergence between the encoder's distribution and a prior distribution, typically assumed to be a standard multivariate Gaussian. The regularization term ensures the latent space has good properties, encouraging the model to use all dimensions of the latent space and avoid overfitting.

In the version of VAEs as detailed in Dai and Wipf (2019), a trainable Gamma parameter is introduced in the reconstruction loss term, which we follow in this study. This Gamma parameter effectively controls the balance between the reconstruction loss and the KL divergence. By learning a suitable value for this parameter, the VAE is better equipped to capture the structure of the data and adjust the strength of the regularization based on the complexity of the dataset, potentially leading to more robust and interpretable results. A conditional VAE simply adds conditioning information to the encoder and decoder, to allow the distribution to depend on some additional factor, in this case, the trophic niche of the birds.

Fitting this secondary condition VAE allowed us to incorporate some uncertainty into the generative process, as well as estimate a lower-dimensional manifold embedded in the latent space on which the bird beaks seemed to lie. It also dealt with a common problem in generative models, known as the 'prior hole' problem, which can also be an issue in VAEs [45,46]. Generally speaking this problem is caused by the training process encouraging conformity to a prior distribution but being unable to fully enforce it. This means the estimated latent vectors of the data (or 'aggregate posterior') typically do not match the prior and thus are not fully independently Gaussian. This becomes a problem for generation because generation usually involves sampling a latent vector from the prior distribution and then running this through the decoder. If the estimated latent vectors deviate from the prior substantially then the generated samples will not be representative. Indeed, in our case, we found the latent vectors estimated for the bird beaks deviated substantially from an independent Gaussian, specifically most latent dimensions showed substantially thicker tails than expected under a Gaussian distribution (S1A Fig). However Dai and Wipf (2019) showed that fitting a second VAE on the results of a first-stage VAE resulted in much better conformity to the desired prior. Here we find that this is also the case when fitting a VAE on the results of our autodecoder architecture, the second-stage VAE estimated latent space conformed strongly to an independent Gaussian distribution (S1B Fig). This allowed us to create a model where we can generate hypothetical samples of bird beaks by sampling latent vectors from an independent Gaussian distribution, running them through the VAE decoder, then running the results of this, in turn, through the DeepSDF decoder to get the final bird beak mesh.

We additionally made the second-stage VAE a conditional VAE (CVAE), that was conditioned on the trophic niche of the bird to which each beak belonged. This allows the model to generate from the conditional distribution of bird beaks given a trophic niche, which was is complimentary to our Trophic Niche classifier analysis (see Downstream Analysis: Trophic Niche Prediction section for details).

## Downstream analysis

**Latent space visualization.** Because the latent morphology space estimated in the model is 64 dimensional, it is difficult to visualize. To visualize the latent morphology space we reduced the dimension of the latent space to two dimensions using commonly used machine learning

algorithms. First, we produced an animation to explore the latent space by first reducing the dimension to two using the t-Distributed Stochastic Neighbor Embedding (t-SNE) algorithm [47] and then creating a 'tour' through the reduced dimension latent space. t-SNE's focus on local structure was suitable for this purpose because it allowed for smoother interpolation through the latent space, reducing the incidence of significant gaps due to strong warping of the estimated underlying manifold. We used the implementation of t-SNE from the sklearn package in Python, and used its default hyper-parameter settings. See S1 Movie for the result.

**Latent vector discovery.** We looked for interpretable 'vectors' in the multidimensional latent space of bird beak morphology by finding the direction in latent space most aligned with independent measurements of the birds beaks. To do this, we collected measurements of length, width (side to side), and depth (top to bottom) of the beaks of bird species from the AVONET dataset [48]. We joined this dataset to the 3D mesh dataset by matching taxonomic names. We calculated two measures of the general shape of the beak, unitless measures we called 'elongation' and 'broadness'. Elongation was defined as the length of the beak (from tip to culmen) divided by the average of the width and depth (measured at the beak base). This measures how long the beak is relative to its 'girth'. Broadness was defined as the the beak's width divided by its depth, which measures how wide the beak is relative to it's 'height'.

We found the vector in latent space most associated with each of these measures using a multivariate regression where the response was either narrowness or flatness and the predictors were the 64 latent variables estimated in the DeepSDF model (with no intercept). The coefficients from this regression represent a vector that points along the plane of best fit in multidimensional space, going from the origin to the maximum point on the plane. Under assumptions of linearity this is the vector of maximum increase with respect to the response. To confirm that these vectors did indeed represent changes along the desired axis of morphological variation, we visualized them. To do so we first created a sample of 10 random bird beaks from the latent space to apply the vectors to. We used 10 random beaks so that we could see if the vectors effects were consistent and independent of the where in the latent space we started from. We sampled vectors of length 64 by drawing from the DeepSDF model prior distribution for the latent space. We then applied the discovered vectors to these 10 random beaks by moving the sampled values along the vectors in both the positive and negative directions. We then reconstructed full 3D shapes from the results and plotted them (Figs 2 and 3).

**Ecological meaning.** We visualized how the morphological latent space aligned with an important ecological factor, the trophic niche of birds, by visualizing another 2 dimensional dimension reduction of the full latent space, and plotting points using different colours for different trophic niches. This time, we employed the Uniform Manifold Approximation and Projection (UMAP) algorithm, a powerful dimensionality reduction technique, to reduce the high-dimensional latent space of bird beak morphology for improved visualization and interpretation [49]. UMAP is a non-linear dimensionality reduction method that excels at preserving both local and global structure of the data while minimizing distortion, making it particularly well-suited for the analysis of complex biological datasets [50]. The UMAP algorithm is founded on the principles of manifold learning and topological data analysis, which seek to approximate and project high-dimensional data onto a lower-dimensional space while maintaining the intrinsic geometry of the data. The algorithm utilizes a combination of fuzzy simplicial sets and spectral embedding to construct a topological representation of the data, enabling the preservation of both local and global structure during dimensionality reduction (Becht et al., 2018). This feature distinguishes UMAP from other dimensionality reduction techniques, such as t-SNE, which primarily focuses on preserving local structure. This makes it more appropriate to look at differences in distinct ecological groups, since it is likely important to take into account the global structure of the data to see where different

groups fall out. We used the UMAP implementation in the R package uwot (with default settings accept for the number of nearest neighbours used for the underlying graph, which we set to 25).

**Trophic niche prediction.** To additionally test the ecological relevance of the estimated beak morphology latent vectors, we trained a Random Forest classifier [51] to predict a bird species' trophic niche from their 64 estimated latent morphology variables. Before training classification models, 89 of the 2,021 datapoints were excluded from the dataset after discovering potential issues in their meshes (mostly badly misaligned lower beaks).

We first removed 20% of the beaks to serve as a hold-out test dataset, with sampling stratified by trophic niche, before training models as described below.

For the training data, 1 out of 10 trophic niche categories were assigned to each bird species using the AVONET dataset [48]. We trained the model using the 'tidymodels' package in R [52]. Overall, trophic niche classes were highly imbalanced in our dataset (Table 1), which is well-known to cause issues in classification models if not dealt with appropriately. We addressed this using case weights in the tidymodels framework. For each observation i in trophic niche class j, we assigned a weight $w_i = 1/n_j$, where $n_j$ is the sample size of class j. This weighting scheme ensures each trophic niche class contributes equally to model training regardless of sample size.

The weights were incorporated using the case_weights() specification in tidymodels, and all model tuning and evaluation maintained these weights through cross-validation. Both Random Forest and multinomial regression models used these same weights, ensuring fair comparison between methods.

We tuned several hyperparameters of the Random Forest model based on a 20 monte carlo cross validation splits, each of which used a random 20% of the training data for model validation. We used the R package tidymodels for model fitting. We tuned the trees, mtry, and min_n parameters, which control the numbers of trees used to build an ensemble, the numbers of variables to include in each individual tree, and the minimum number of data points allowed in a terminal grouping in each tree, respectively.

We also trained a Random Forest classifier on 64 PCA axes calculated from landmarks (same as those used in [23]) after using the same alignment procedure used for the DeepSDF analysis. This provided a direct comparison to our DeepSDF variables by matching dimensionality and preprocessing steps. We used identical case weights and cross-validation procedures as described above.

**Table 1. Sample sizes for different trophic niches in the bird beak dataset. The data is highly imbalanced.**

| Trophic Niche | Sample Size |
| --- | --- |
| Aquatic predator | 197 |
| Frugivore | 211 |
| Granivore | 113 |
| Herbivore aquatic | 20 |
| Herbivore terrestrial | 29 |
| Invertivore | 877 |
| Nectarivore | 130 |
| Omnivore | 381 |
| Scavenger | 13 |
| Vertivore | 50 |

We also ran a regularized multinomial regression for comparison, using the glmnet package through tidymodels. Like our Random Forest models, observations were weighted by the inverse of their class sample size, and elastic net regularization was used with tuned alpha (mixing) and lambda (overall regularization strength) parameters.

All models were evaluated on the original held-out test dataset using accuracy and balanced accuracy as metrics.

**Phylogenetic analysis of morphological variables.** We used the geomorph R package [24] to calculate univariate and multivariate Blomberg's K for three variable sets:

1. 64 PCA axes from landmarks aligned using our DeepSDF preprocessing

2. These PCA axes after phylogenetic PCA redistribution (e.g., using phylogenetic PCA)

3. The 64 DeepSDF latent variables

We also performed Phylogenetically Aligned Components Analysis (PACA) [25] on both PCA and DeepSDF variables to find directions maximizing alignment with phylogenetic signal.

To test whether phylogenetic signal explained predictive power for trophic niche, we repeated trophic niche classification using the PACA aligned variables, but removing the first axis from both representations. The idea of this analysis is that the first PACA axis is the most phylogenetically informative -- by removing it the classification must rely on the remaining axes of variation, which should capture variation less confounded with phylogeny. This helped evaluate whether trophic niche prediction relied primarily on phylogenetically structured variation, or is using information beyond phylogeny.

## Supporting information

**S1 Fig. Histograms and QQ-plots for all 64 latent dimensions estimated by the DeepSDF model (pages 1-8) followed by the same for the 15 latent dimensions of the stage 2 Variational Autoencoder (VAE) subsequently trained on the original 64 dimensions (pages 9-10).** The stage 2 VAE histograms and QQ-plots show distinctively increased conformation to Gaussian distributions.
(PDF)

**S1 Movie. Exploring the bird beak morphological latent space.** The animation shows a tour through the latent space of the model. For visualization purposes the 64 dimensional space of the model was reduced to two using t-sne on the latent codes of the observed bird beaks. Red filled circles correspond to real bird beaks observed in the dataset. The tour was constructed by choosing a set of random species from the dataset (larger red circles) and then linking them with a cubic spline (shown as a black line), in both the original 64 dimensional space and the two dimensonal t-SNE space.
(GIF)

## Acknowledgments

We'd like to thank Scott Edwards, Janna Fierst, Jason Pienaar, and their labs for thoughtful comments and discussion on the model presented in this study.

## Author contributions

**Conceptualization:** Russell Dinnage, Marian Kleineberg.

**Data curation:** Russell Dinnage.

**Formal analysis:** Russell Dinnage, Marian Kleineberg.

**Investigation:** Russell Dinnage, Marian Kleineberg.

**Methodology:** Russell Dinnage, Marian Kleineberg.

**Project administration:** Russell Dinnage.

**Resources:** Russell Dinnage, Marian Kleineberg.

**Software:** Russell Dinnage, Marian Kleineberg.

**Validation:** Russell Dinnage.

**Visualization:** Russell Dinnage, Marian Kleineberg.

**Writing – original draft:** Russell Dinnage.

**Writing – review & editing:** Russell Dinnage, Marian Kleineberg.

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
