## [Decision Letter · Decision Letter 0]

30 Oct 2023

Dear Dr. Dinnage,

Thank you very much for submitting your manuscript "Generative A.I. helps extract ecological meaning from the complex three dimensional shapes of bird bills" for consideration at PLOS Computational Biology.

As with all papers reviewed by the journal, your manuscript was reviewed by members of the editorial board and by several independent reviewers. In light of the reviews (below this email), we would like to invite the resubmission of a significantly-revised version that takes into account the reviewers' comments.

I found this a very interesting study, and all three reviewers agree. However, the reviewers raised important issues which must be addressed before a decision can be made. As Reviewers #1 and #3 point out, it is important to discuss the accuracy and overall performance of the DeepSDF method on trophic niche predictions relative to more established 3D morphometric techniques. Also needing more careful discussion, as noted by Reviewer #2, is the overfitting in the trophic niche analysis suggested by the drop from 100% to 62 % between the training and test data. This feeds into the concerns raised by Reviewer #3 about the generalizability of this method to potentially much smaller datasets and other taxa. 

We cannot make any decision about publication until we have seen the revised manuscript and your response to each of the reviewers' comments. Your revised manuscript is also likely to be sent to reviewers for further evaluation.

Sincerely,

Rafael D'Andrea, Ph.D.

Academic Editor

PLOS Computational Biology

Feilim Mac Gabhann

Editor-in-Chief

PLOS Computational Biology

Reviewer's Responses to Questions

**Comments to the Authors:**

Reviewer #1: This paper reports on an AI-based approach for analysing complex shape variation in 3-d scans of bird beaks. The study employs an ‘autodecoder’ architecture combined with a subsequent variational autoencoder (VAE) step to first learn and then generate coherent representations of bill shape variation. The authors go on to show that the latent vectors learnt by the model capture intuitive aspects of bill shape variation (‘elongation’, ‘broadness’) and can be used to predict variation in species feeding ecology with some accuracy.

Overall I think this is a valuable study that describes a potentially important new approach for extracting meaningful representations of complex 3-d morphological datasets such as this. The manuscript is generally clearly written and well-presented and the visualisations are useful and impressive. That said, from the perspective of a comparative biologist with experience of working with 3-d bill shape data from museum specimens I have some queries/suggestions/concerns to consider:

1. As mentioned, from experience I am aware that the positioning of the lower bill relative to the upper bill can vary idiosyncratically bird museum specimens as a result of preservation style, damage etc. The method here uses the whole bill scan as input and therefore this unhelpful variation in relative bill position is presumably encoded into the latent vector representation and may be problematic. Are the authors able to probe the extent to which variation in the relative positioning of the upper and lower beak is encoded by the model and whether this is problematic for downstream analysis? For example, could the model be retrained with only the top bill included as input and the results compared? I raise this as the original study to analyse the beak shape data (Cooney et al. 2017, Nature) focused only on the shape of the upper bill to avoid this issue.

2. Regarding the latent vector discovery analysis (starting line 304), as well as the AVONET measurements, why not also use the principal component (PC) scores of 3-d bill shape generated in the original paper (Cooney et al. 2017, Nature) to probe for meaningful vectors? The Cooney et al. PC scores have the benefit of being measured on the precisely the same dataset as used by the present study and therefore should be more comparable (though less independent) than the AVONET linear measurements. These PC scores also capture more complex shape variation than is possible from the more simple length, width depth measurements, such as ‘degree of hookiness’ alluded to in the Discussion (line 538).

3. I think it the discussion of the tropic niche results needs to be extensively revised to incorporate the findings of Pigot et al. (2020) (https://doi.org/10.1038/s41559-019-1070-4), who performed a very similar classification analysis using not only beak shape data but also other morphological measurements (e.g. leg morphology) as proposed in the Discussion (line 471). Currently Pigot et al. (2020) is not cited and is conspicuous by its absence.

4. Regarding the trophic niche analysis, in my view it would be highly beneficial to compare the prediction accuracy of the latent vectors generated here to that based on other beak shape datasets. In particular, it would be informative to contrast the performance of the latent vectors against the predictive accuracy of (i) the PCs scores from the original paper (Cooney et al. 2017) and (ii) the linear beak shape measurements from AVONET (or Pigot et al. 2020). Currently, the authors argue that their latent vector representations are ‘ecologically meaningful’, but it’s not clear how meaningful they are relative to alternative

more established techniques based on linear measurements and 3-d morphometrics.

Minor:

L299 – what parameter settings were used to generate the t-SNE visualisation?

L329 – what parameter settings were used to generate the UMAP visualisation?

Figure 4 – It would be useful perhaps to add an additional panel to this figure with datapoints coloured by taxonomic group (e.g. order). This would help provide insight into the composition of ‘outlier’ groups (e.g. points in the top left of fig. 4a), specifically whether they constitute sets of related species or independently evolved convergent morphologies.

Reviewer #2: In their manuscript, "Generative A.I. helps extract ecological meaning from the complex three dimensional shapes of bird bills", Dinnage & Kleineberg train a deep representational learning model on 2000+ 3D scans of bird beaks from the public MarkMyBird dataset. The learned latent space is further associated with measures of beak elongation and broadness, showing that ecologically relevant morphological variation is represented by the learned latent space. The model is also able to predict the trophic niche associated with a particular beak shape.

The study represents an interesting and novel application of generative A.I. models to the study of 3D morphological shape and correlations with ecological roles. In general, the manuscript is also well-written and clear, though there are some sections that would benefit from more explicit quantification (see below).

The most major issue I identified is that it appears that the trophic niche prediction model is overfitted, given the very high accuracy (100%) on the training set but comparatively poor performance on the test set (64%). This suggests that the model has basically memorized the training set but has quite poor generalization capabilities. While the authors make some arguments as to why the results are valid despite the relatively low test accuracy, they never explicitly address this problem of overfitting in the Results or Discussion (the word 'overfitting' is only mentioned with respect to regularization in the Methods [line 287], and only in a general sense). This omission needs to be corrected as the model, based on what is presented, is very clearly overfitted and this is a fundamental issue in these types of learning problems. What, if anything, was done to try to fix the overfitting problem? Is the model too complex, are there too many learning parameters, is there not enough data, etc.? Some of these questions are answered (or implied) in other parts of the paper, but I think there needs to be a focused discussion with specific reference to the fact that the model is overfitted.

With regard to Figure 5, I think it would help interpretability if: (1) The # of samples per class is marked for each trophic niche (on both the left and the right). The current "freq" axis is not very useful because it is spread across all classes; (2) The % match is shown on the left side of the plot -- for example, 80% match in prediction for Invertivore class, etc. These numbers should also be folded into the discussion in the text (lines 392-406), which is too qualitative at the moment, relying on ambiguous qualifiers such as "most", "more", "sometimes", etc. The class labels should also be out to the side instead of overlapping the flows for the sake of visibility.

Minor Comments

Line 128-131: This sentence is confusingly constructed and hard to understand; consider rewriting as multiple sentences.

Line 198: "conditional" not "confitional"

I think it would make sense to put the section describing VAEs (lines 270-293) earlier than where it is now, as currently the manuscript discusses VAEs before you define what they are. Towards the beginning of the Trophic Niche Conditional VAE section would probably be best (rather than at the end, as now).

Line 348: "well known" not "well know"

The authors should provide a table or list of the number of samples per trophic niche when discussing the class imbalance issue (lines 347-357).

Line 358: There is an extra space after "min_n parameters"

Line 385: "trophic" not "tropich"

Line 397-400: This is a run-on sentence.

Reviewer #3: The review was uploaded as an attachment.

**Have the authors made all data and (if applicable) computational code underlying the findings in their manuscript fully available?**

Reviewer #1: Yes

Reviewer #2: Yes

Reviewer #3: Yes

PLOS authors have the option to publish the peer review history of their article (what does this mean? ). If published, this will include your full peer review and any attached files.

**Do you want your identity to be public for this peer review?** For information about this choice, including consent withdrawal, please see our Privacy Policy .

Reviewer #1: No

Reviewer #2: No

Reviewer #3: No
---

## [Decision Letter · Decision Letter 1]

11 Jul 2024

Dear Dr. Dinnage,

Thank you very much for submitting your manuscript "Generative AI helps extract ecological meaning from the complex three dimensional shapes of bird bills" for consideration at PLOS Computational Biology.

My apologies for the slow review process. It has proven difficult to find reviewers for this paper after two of the original reviewers became unavailable. One reviewer of the original manuscript, one new reviewer, and I have now assessed the revised manuscript. I appreciate the effort put into this revision, which in my view has substantially improved the manuscript. Reviewer #1 is satisfied that the revision fully addressed their comments. In my judgment, the comments of Reviewer #2 and most of those raised by Reviewer #3 have also been successfully addressed. However, some issues remain in relation to performance, assumptions, and potential disadvantages of DeepSDF relative to other methods already in use, and these concerns are echoed and expanded upon by Reviewer #4. Specifically, concerns were raised about the robustness and interpretability of the latent space, the potential phylogenetic structure of the data, and the appropriateness of labeling DeepSDF as a foundational model. I believe the authors will find it relatively straightforward to address these remaining issues in a second revision.

We cannot make any decision about publication until we have seen the revised manuscript and your response to the reviewers' comments. Your revised manuscript is also likely to be sent to reviewers for further evaluation.

Sincerely,

Rafael D'Andrea, Ph.D.

Academic Editor

PLOS Computational Biology

Feilim Mac Gabhann

Editor-in-Chief

PLOS Computational Biology

Reviewer's Responses to Questions

**Comments to the Authors:**

Reviewer #1: I thank the authors for engaging with my initial set of comments on their manuscript. Overall I feel that the authors have addressed my points satisfactorily, and I particularly welcome the additional analysis investigating the lower beak issue and the comparison of results to PC scores. Overall I think the revised manuscript is much improved compared to the original and I have no further points to raise. I think this is an interesting and valuable study that will be of great interest to comparative biologists working with complex 3D data.

Reviewer #4: In this manuscript the authors apply the neural network, DeepSDF, as a tool for non-linear dimension reduction. The authors’ aim to use the estimated latent space as a tool for downstream analysis such as ancestral state reconstruction and predicting ecological associations. The major advancement of this paper is that it would give biologists a tool other than PCA to analyze complex shape such as those depicted by 3D meshes or point clouds. On the one hand, I believe this is an interesting contribution to the field and would generate discussion on the topic of shape evolution. There is an intuitive appeal of reducing the dimensionality of shape in a non-linear way and many of the AI topics (e.g., manifold hypothesis) discussed in this paper would be useful for biologists to consider. On the other hand, I don’t believe the authors have been nearly skeptical enough about the DeepSDF method. As other reviewers have pointed out, there are times when the statements made in this manuscript do not seem to be backed up by the results. Many of the assessments being made are qualitative and the caveats and assumptions of this method are not clearly laid out. My suggestion is that the authors tone down their language when discussing this work and consider it with a more critical eye. Below I will outline some of the claims I believe the authors have taken too far.

One of my primary concerns with this paper is the claim that non-linear dimension reduction represents a significant step forward for complex shape evolution when essentially none of the results support that claim. Unless I misunderstood, it seemed that PCA generally preformed better than or equal to DeepSDF across most tests. So, other than the somewhat intuitive idea that shape is non-linear and therefore our dimension reduction should be as well, I don’t believe the authors provide convincing evidence that DeepSDF is the ideal way to model shape.

Furthermore, the authors do not mention several ways in which PCA may be preferable to non-linear dimension reduction. I believe the manuscript generally treats the problem in a one-sided way in which the non-linear techniques are always going to be better than linear counterparts. However, this is based on intuition rather than any theoretical prediction. Below, I will outline 3 ways a PCA would be preferable to a non-linear technique.

1) The authors can correct me if I’m wrong, but generally these deep neural networks are not deterministic in the way that they model the latent space. This means that each time the model is run, there is potential for the latent space to be different. Besides the issues of reproducibility, this is potentially problematic if the latent spaces differ significantly each time the model is trained. How robust is this latent space simply to stochasticity in training? Are distances between points maintained when the model is trained? These distances will impact rate estimation and therefore reconstructions, so this is an important point to understand.

2) What exactly is being modeled in the latent space. One of the conveniences of PCA is that the principal components are generally easily defined by the input features. This is more complex for shape data, but it should be true that we can get insight into the latent PC space numerically and have an idea of the amount of variation explained when reducing the dimensionality. Is this possible for the latent space produced by an autoencoder? Is there any way to determine how much information is being lost upon dimension reduction? Is there anyway to understand the latent dimensions other than qualitative assessments of the reconstructions from the latent space? My concern is that we do not know what that latent space is and to model it as though it represents a real biological measurement could be dubious.

3) PCA has methods for phylogenetic correction which allow for the partitioning of variation due to common ancestry and variation in morphology. I realize the authors discuss future plans to extend this work with a phylogenetic component, but there is no mention of this assumption prior to that point in the discussion. How do we know that the latent space presented is not simply a reflection of the phylogeny? If the latent space is phylogenetically structured, then results related to ecological association could be due to that rather than the morphology - all that would be needed is that ecology is also phylogenetically structured. This seems to be a major caveat of the method.

Another claim which I think needs to be addressed is the classification of this as a foundation model. The concept of foundation models lacks a clear definition, probably because it was essentially a corporate marketing strategy. Nonetheless, so-called “foundation” models do serve as a basis for many fine-tuned applications and are incredibly useful for AI applications. In contrast to the training in this paper, foundation models are trained on extensive datasets (millions of samples if not more), often carefully curated to ensure wide-ranging applicability. In my opinion, 2000 samples is not enough for this model to be labeled as foundational. For example, it is unlikely that this model could be applied effectively to mammals skulls, reptile skulls, or any 3D organismal representations, as these examples seem to fall outside the model's distribution. It seems unlikely that this model generally understands and can model “3D morphology.” This is where the issue of the lack of definition of “foundational” model is problematic, since this model may preform well on beak data generally. But is that enough to claim foundation model? If we consider foundation models from AI generally, we see that they can perform tasks for particular data types well. For example, SegmentAnything can segment any model instantly without additional training and fine-tuning that model has led to many interesting developments within AI. Foundational language models like ChatGPT excel in diverse language tasks. The authors have not demonstrated the generality of their method beyond their dataset, making the claim of it being a foundational model an overreach.

Line 169: “We also trained a generalized linear model (with regularization) using the glmnet R package using both our…”

In my copy of the manuscript, this is where the paragraph ends. Clearly something has been cut-off.

Line 313: “potential is enormous”. Based on what evidence do you make this claim. Deep learning is of course an incredibly powerful tool. But the model presented here was only able to match PCA. Would you say that PCA has enormous potential?

**Have the authors made all data and (if applicable) computational code underlying the findings in their manuscript fully available?**

Reviewer #1: Yes

Reviewer #4: Yes

PLOS authors have the option to publish the peer review history of their article (what does this mean? ). If published, this will include your full peer review and any attached files.

**Do you want your identity to be public for this peer review?** For information about this choice, including consent withdrawal, please see our Privacy Policy .

Reviewer #1: No

Reviewer #4: No
---

## [Decision Letter · Decision Letter 2]

20 Jan 2025

PCOMPBIOL-D-23-01064R2

Generative AI helps extract ecological meaning from the complex three dimensional shapes of bird bills

PLOS Computational Biology

Dear Dr. Dinnage,

Thank you for submitting your manuscript to PLOS Computational Biology. After careful consideration, we feel that it has merit but does not fully meet PLOS Computational Biology's publication criteria as it currently stands. Therefore, we invite you to submit a revised version of the manuscript that addresses the points raised during the review process.

Please submit your revised manuscript within 30 days. If you will need more time than this to complete your revisions, please reply to this message or contact the journal office at ploscompbiol@plos.org. Please include the following items when submitting your revised manuscript:

We look forward to receiving your revised manuscript.

Kind regards,

Rafael D'Andrea, Ph.D.

Academic Editor

PLOS Computational Biology

Feilim Mac Gabhann

Editor-in-Chief

PLOS Computational Biology

**Journal Requirements:**

**Reviewers' comments:**

Reviewer's Responses to Questions

**Comments to the Authors:**

Reviewer #4: I thank the authors for their thorough response to my previous comments. I believe the more detailed PCA analysis and examination of phylogenetic signal represent a step forward in our understanding of non-linear latent spaces in the context of phylogenetic relationships.

My only concern remains classifying this as the first morphological "foundation model." The amount of data used to train a foundation model is many orders of magnitude greater than what was used in this study. For example, the recent sequence foundation model (Nguyen et al. 2024) and ct foundation model (Google research 2024) were trained on 2.7 million sequences and 3.7 million image scans respectively. These are foundation models because their extensive training allows the model to potentially gain some higher level understanding of their task. This higher level understanding then allows the foundation models to preform zero-shot learning and be used as a base for future fine tuning. 2,000 data samples is simply not enough for this model to be classified as a foundation model. The authors should remove claims that this is a foundation model. It is not necessary to include in this manuscript and, in my opinion, it is a false claim.

Nonetheless, this is an exciting proof of concept! With much additional training data, this model could prove to be a foundation model (though, in truth, I suspect a more complex architecture will be necessary to take advantage of larger datasets). And other than the claim of DeepSDF as a foundation model, I have no issues with the manuscript as is. I am excited to see this work progress and the advances that can be made in this area.

**Have the authors made all data and (if applicable) computational code underlying the findings in their manuscript fully available?**

Reviewer #4: Yes

PLOS authors have the option to publish the peer review history of their article (what does this mean? ). If published, this will include your full peer review and any attached files.

**Do you want your identity to be public for this peer review?** For information about this choice, including consent withdrawal, please see our Privacy Policy .

Reviewer #4: No

**Figure resubmission:**
---

## [Editor Report · Decision Letter 3]

18 Feb 2025

Dear Dr. Dinnage,

We are pleased to inform you that your manuscript 'Generative AI helps extract ecological meaning from the complex three dimensional shapes of bird bills' has been provisionally accepted for publication in PLOS Computational Biology.

Best regards,

Rafael D'Andrea, Ph.D.

Academic Editor

PLOS Computational Biology

Feilim Mac Gabhann

Editor-in-Chief

PLOS Computational Biology

---

## [Editor Report · Acceptance letter]

PCOMPBIOL-D-23-01064R3

Generative AI helps extract ecological meaning from the complex three dimensional shapes of bird bills

Dear Dr Dinnage,

I am pleased to inform you that your manuscript has been formally accepted for publication in PLOS Computational Biology. Your manuscript is now with our production department and you will be notified of the publication date in due course.

With kind regards,

Anita Estes
